# Dynamic matrix recovery from incomplete observations under an exact low-rank constraint

**Liangbei Xu     Mark A. Davenport**
Department of Electrical and Computer Engineering
Georgia Institute of Technology
Atlanta, GA 30318
lxu66@gatech.edu     mdav@gatech.edu

## Abstract

Low-rank matrix factorizations arise in a wide variety of applications – including recommendation systems, topic models, and source separation, to name just a few. In these and many other applications, it has been widely noted that by incorporating temporal information and allowing for the possibility of time-varying models, significant improvements are possible in practice. However, despite the reported superior empirical performance of these dynamic models over their static counterparts, there is limited theoretical justification for introducing these more complex models. In this paper we aim to address this gap by studying the problem of recovering a dynamically evolving low-rank matrix from incomplete observations. First, we propose the locally weighted matrix smoothing (LOWEMS) framework as one possible approach to dynamic matrix recovery. We then establish error bounds for LOWEMS in both the *matrix sensing* and *matrix completion* observation models. Our results quantify the potential benefits of exploiting dynamic constraints both in terms of recovery accuracy and sample complexity. To illustrate these benefits we provide both synthetic and real-world experimental results.

## 1   Introduction

Suppose that $X \in \mathbb{R}^{n_1 \times n_2}$ is a rank-$r$ matrix with $r$ much smaller than $n_1$ and $n_2$. We observe $X$ through a linear operator $\mathcal{A} : \mathbb{R}^{n_1 \times n_2} \to \mathbb{R}^m$,

$$y = \mathcal{A}(X), \quad y \in \mathbb{R}^m.$$

In recent years there has been a significant amount of progress in our understanding of how to recover $X$ from observations of this form even when the number of observations $m$ is much less than the number of entries in $X$. (See [8] for an overview of this literature.) When $\mathcal{A}$ is a set of weighted linear combinations of the entries of $X$, this problem is often referred to as the *matrix sensing* problem. In the special case where $\mathcal{A}$ samples a subset of entries of $X$, it is known as the *matrix completion* problem. There are a number of ways to establish recovery guarantee in these settings. Perhaps the most popular approach for theoretical analysis in recent years has focused on the use of nuclear norm minimization as a convex surrogate for the (nonconvex) rank constraint [1, 3, 4, 5, 6, 7, 15, 19, 21, 22]. An alternative, however is to aim to directly solve the problem under an exact low-rank constraint. This leads a non-convex optimization problem, but has several computational advantages over most approaches to minimizing the nuclear norm and is widely used in large-scale applications (such as recommendation systems) [16]. In general, popular algorithms for solving the rank-constrained models – e.g., alternating minimization and alternating gradient descent – do not have as strong of convergence or recovery error guarantees due to the non-convexity of the rank constraint. However, there has been significant progress on this front in recent years [11, 10, 12, 13, 14, 23, 25], with many of these algorithms now having guarantees comparable to those for nuclear norm minimization.

Nearly all of this existing work assumes that the underlying low-rank matrix $X$ remains fixed throughout the measurement process. In many practical applications, this is a tremendous limitation. For example, users' preferences for various items may change (sometimes quite dramatically) over time. Modelling such drift of user's preference has been proposed in the context of both music and movies as a way to achieve higher accuracy in recommendation systems [9, 17]. Another example in signal processing is dynamic non-negative matrix factorization for the blind signal separation problem [18]. In these and many other applications, explicitly modelling the dynamic structure in the data has led to superior empirical performance. However, our theoretical understanding of dynamic low-rank matrix recovery is still very limited.

In this paper we provide the first theoretical results on the dynamic low-rank matrix recovery problem. We determine the sense in which dynamic constraints can help to recover the underlying time-varying low-rank matrix in a particular dynamic model and quantify this impact through recovery error bounds. To describe our approach, we consider a simple example where we have two rank-$r$ matrices $X^1$ and $X^2$. Suppose that we have a set of observations for each of $X^1$ and $X^2$, given by

$$y^i = \mathcal{A}^i\left(X^i\right), \quad i = 1, 2.$$

The naïve approach is to use $y^1$ to recover $X^1$ and $y^2$ to recover $X^2$ separately. In this case the number of observations required to guarantee successful recovery is roughly $m^i \geq C^i r \max(n_1, n_2)$ for $i = 1, 2$ respectively, where $C^1, C^2$ are fixed positive constants (see [4]). However, if we know that $X^2$ is close to $X^1$ in some sense (for example, if $X^2$ is a small perturbation of $X^1$), then the above approach is suboptimal both in terms of recovery accuracy and sample complexity, since in this setting $y^1$ actually contains information about $X^2$ (and similarly, $y^2$ contains information about $X^1$). There are a variety of possible approaches to incorporating this additional information. The approach we will take is inspired by the LOWESS (locally weighted scatterplot smoothing) approach from non-parametric regression. In the case of this simple example, if we look just at the problem of estimating $X^2$, our approach reduces to solving a problem of the form

$$\min_{X^2} \|\mathcal{A}^2(X^2) - y^2\|_2^2 + \lambda \|\mathcal{A}^1(X^2) - y^1\|_2^2 \qquad \text{s.t.} \quad \text{rank}\left(X^2\right) \leq r,$$

where $\lambda$ is a parameter that determines how strictly we are enforcing the dynamic constraint (if $X^1$ is very close to $X^2$ we can set $\lambda$ to be larger, but if $X^1$ is far from $X^2$ we will set it to be comparatively small). This approach generalizes naturally to the *locally weighted matrix smoothing* (LOWEMS) program described in Section 2. Note that it has a (simple) convex objective function, but a non-convex rank constraint. Our analysis in Section 3 shows that the proposed program outperforms the above naïve recovery strategy both in terms of recovery accuracy and sample complexity.

We should emphasize that the proposed LOWEMS program is non-convex due to the exact low-rank constraint. Inspired by previous work on matrix factorization, we propose using an efficient alternating minimization algorithm (described in more detail in Section 4). We explicitly enforce the low-rank constraint by optimizing over a rank-$r$ factorization and alternately minimize with respect to one of the factors while holding the other one fixed. This approach is popular in practice since it is typically less computationally complex than nuclear norm minimization based algorithms. In addition, thanks to recent work on global convergence guarantees for alternating minimization for low-rank matrix recovery [10, 13, 25], one can reasonably expect similar convergence guarantees to hold for alternating minimization in the context of LOWEMS, although we leave the pursuit of such guarantees for future work.

To empirically verify our analysis, we perform both synthetic and real world experiments, described in Section 5. The synthetic experimental results demonstrate that LOWEMS outperforms the naïve approach in practice both in terms of recovery accuracy and sample complexity. We also demonstrate the effectiveness of LOWEMS in the context of recommendation systems.

Before proceeding, we briefly state some of the notation that we will use throughout. For a vector $x \in \mathbb{R}^n$, we let $\|x\|_p$ denote the standard $\ell_p$ norm. Given a matrix $X \in \mathbb{R}^{n_1 \times n_2}$, we use $X_{i:}$ to denote the $i^{\text{th}}$ row of $X$ and $X_{:j}$ to denote the $j^{\text{th}}$ column of $X$. We let $\|X\|_F$ denote the the Frobenius norm, $\|X\|_2$ the operator norm, $\|X\|_*$ the nuclear norm, and $\|X\|_\infty = \max_{i,j} |X_{ij}|$ the element-wise infinity norm. Given a pair of matrices $X, Y \in \mathbb{R}^{n_1 \times n_2}$, we let $\langle X, Y \rangle = \sum_{i,j} X_{ij} Y_{ij} = \text{Tr}\left(Y^T X\right)$ denote the standard inner product. Finally, we let $n_{\max}$ and $n_{\min}$ denote $\max\{n_1, n_2\}$ and $\min\{n_1, n_2\}$ respectively.

## 2 Problem formulation

The underlying assumption throughout this paper is that our low-rank matrix is changing over time during the measurement process. For simplicity we will model this through the following discrete dynamic process: at time $t$, we have a low-rank matrix $X^t \in \mathbb{R}^{n_1 \times n_2}$ with rank $r$, which we assume is related to the matrix at previous time-steps via

$$X^t = f(X^1, \ldots, X^{t-1}) + \epsilon^t,$$

where $\epsilon^t$ represents noise. Then we observe each $X^t$ through a linear operator $\mathcal{A}^t : \mathbb{R}^{n_1 \times n_2} \to \mathbb{R}^{m^t}$,

$$y^t = \mathcal{A}^t(X^t) + z^t, \quad y^t, z^t \in \mathbb{R}^{m^t}, \tag{1}$$

where $z^t$ is measurement noise. In our problem we will suppose that we observe up to $d$ time steps, and our goal is to recover $\{X^t\}_{t=1}^d$ jointly from $\{y^t\}_{t=1}^d$.

The above model is sufficiently flexible to incorporate a wide variety of dynamics, but we will make several simplifications. First, we note that we can impose the low-rank constraint explicitly by factorizing $X^t$ as $X^t = U^t (V^t)^T, U^t \in \mathbb{R}^{n_1 \times r}, V^t \in \mathbb{R}^{n_2 \times r}$. In general both $U^t$ and $V^t$ may be changing over time. However, in some applications, it is reasonable to assume that only one set of factors is changing. For example, in a recommendation system where our matrix represent user preferences, if the rows correspond to items and the columns correspond to users, then $U^t$ contains the latent properties of the items and $V^t$ models the latent preferences of the users. In this context it is reasonable to assume that only $V^t$ changes over time [9, 17], and that there is a fixed matrix $U$ (which we may assume to be orthonormal) such that we can write $X^t = UV^t$ for all $t$. Similar arguments can be made in a variety of other applications, including personalized learning systems, blind signal separation, and more.

Second, we assume a Markov property on $f$, so that $X^t$ (or equivalently, $V^t$) only depends on the previous $X^{t-1}$ (or $V^{t-1}$). Furthermore, although other dynamic models could be accommodated, for the sake of simplicity in our analysis we consider the simple model on $V^t$ where

$$V^t = V^{t-1} + \epsilon^t, \quad t = 2, \ldots, d. \tag{2}$$

We will also assume that both $\epsilon^t$ and the measurement noise $z^t$ are i.i.d. zero-mean Gaussian noise.

To simplify our discussion, we will assume that our goal is to recover the matrix at the most recent time-step, i.e., we wish to estimate $X^d$ from $\{y^t\}_{t=1}^d$. Our general approach can be stated as follows. The LOWEMS estimator is given by the following optimization program:

$$\hat{X}^d = \underset{X \in \mathbb{C}(r)}{\arg \min} \mathcal{L}(X) = \underset{X \in \mathbb{C}(r)}{\arg \min} \frac{1}{2} \sum_{t=1}^d w_t \left\| \mathcal{A}^t(X) - y^t \right\|_2^2, \tag{3}$$

where $\mathbb{C}(r) = \{X \in \mathbb{R}^{n_1 \times n_2} : \text{rank}(X) \leq r\}$, and $\{w_t\}_{t=1}^d$ are non-negative weights. We further assume $\sum_{t=1}^d w_t = 1$ to avoid ambiguity. In the following section we provide bounds on the performance of the LOWEMS estimator for two common choices of operators $\mathcal{A}^t$.

## 3 Recovery error bounds

Given the estimator $\hat{X}^d$ from (3), we define the recovery error to be $\Delta^d := \hat{X}^d - X^d$. Our goal in this section will be to provide bounds on $\|\hat{X}^d - X^d\|_F$ under two common observation models. Our analysis builds on the following (deterministic) inequality.

**Proposition 3.1.** *Both the estimator $\hat{X}^d$ by (3) and (9) satisfies*

$$\sum_{t=1}^d w_t \left\| \mathcal{A}^t(\Delta^d) \right\|_2^2 \leq 2\sqrt{2r} \left\| \sum_{t=1}^d w_t \mathcal{A}^{t*}(h^t - z^t) \right\|_2 \left\| \Delta^d \right\|_F, \tag{4}$$

*where $h^t = \mathcal{A}^t(X^d - X^t)$ and $\mathcal{A}^{t*}$ is the adjoint operator of $\mathcal{A}^t$.*

This is a deterministic result that holds for any set of $\{\mathcal{A}^t\}$. The remaining work is to lower bound the LHS of (4), and upper bound the RHS of (4) for concrete choices of $\{\mathcal{A}^t\}$. In the following sections we derive such bounds in the settings of both Gaussian matrix sensing and matrix completion. For simplicity and without loss of generality, we will assume $m^1 = \ldots = m^d =: m_0$, so that the total number of observations is simply $m = dm_0$.

## 3.1 Matrix sensing setting

For the matrix sensing problem, we will consider the case where all operators $\mathcal{A}^t$ correspond to Gaussian measurement ensembles, defined as follows.

**Definition 3.2.** [4] A linear operator $\mathcal{A} : \mathbb{R}^{n_1 \times n_2} \to \mathbb{R}^m$ is a Gaussian measurement ensemble if we can express each entry of $\mathcal{A}(X)$ as $[\mathcal{A}(X)]_i = \langle A_i, X \rangle$ for a matrix $A_i$ whose entries are i.i.d. according to $\mathcal{N}(0, 1/m)$, and where the matrices $A_1, \ldots, A_m$ are independent from each other.

Also, we define the matrix restricted isometry property (RIP) for a linear map $\mathcal{A}$.

**Definition 3.3.** [4] For each integer $r = 1, \ldots, n_{\min}$, the isometry constant $\delta_r$ of $\mathcal{A}$ is the smallest quantity such that

$$(1 - \delta_r) \|X\|_F^2 \leq \|\mathcal{A}(X)\|_2^2 \leq (1 + \delta_r) \|X\|_F^2$$

holds for all matrices $X$ of rank at most $r$.

An important result (that we use in the proof of Theorem 3.4) is that Gaussian measurement ensembles satisfy the matrix RIP with high probability provided $m \geq C r n_{\max}$. See, for example, [4] for details.

To obtain an error bound in the matrix sensing case we lower bound the LHS of (4) using the matrix RIP and upper bound the stochastic error (the RHS of (4)) using a covering argument. The following is our main result in the context of matrix setting.

**Theorem 3.4.** *Suppose that we are given measurements as in* (1) *where all $\mathcal{A}^t$'s are Gaussian measurement ensembles. Assume that $X^t$ evolves according to* (2) *and has rank $r$. Further assume that the measurement noise $z^t$ is i.i.d. $\mathcal{N}(0, \sigma_1^2)$ for $1 \leq t \leq d$ and that the perturbation noise $\epsilon^t$ is i.i.d. $\mathcal{N}(0, \sigma_2^2)$ for $2 \leq t \leq d$. If*

$$m_0 \geq D_1 \max \left\{ n_{\max} r \sum_{t=1}^{d} w_t^2, n_{\max} \right\}, \tag{5}$$

*where $D_1$ is a fixed positive constant, then the estimator $\hat{X}^d$ from* (3) *satisfies*

$$\left\| \Delta^d \right\|_F^2 \leq C_0 \left( \sum_{t=1}^{d} w_t^2 \sigma_1^2 + \sum_{t=1}^{d-1} (d-t) w_t^2 \sigma_2^2 \right) n_{\max} r \tag{6}$$

*with probability at least $P_1 = 1 - d C_1 \exp(-c_1 n_2)$, where $C_0, C_1, c_1$ are positive constants.*

If we choose the weights as $w_d = 1$ and $w_t = 0$ for $1 \leq t \leq d-1$, the bound in Theorem 3.4 reduces to a bound matching classical (static) matrix recovery results (see, for example, [4] Theorem 2.4). Also note that in this case Theorem 3.4 implies exact recovery when the sample complexity is $O(rn/d)$. In order to help interpret this result for other choices of the weights, we note that for a given set of parameters, we can determine the optimal weights that will minimize this bound. Towards this end, we define $\kappa := \sigma_2^2 / \sigma_1^2$ and set $p_t = (d-t), 1 \leq t \leq d$. Then one can calculate the optimal weights by solving the following quadratic program:

$$\{w_t^*\}_{t=1}^{d} = \underset{\sum_t w_t = 1;\ w_t \geq 0}{\arg\min} \sum_{t=1}^{d} w_t^2 + \sum_{t=1}^{d-1} p_t \kappa w_t^2. \tag{7}$$

Using the method of Lagrange multipliers one can show that (7) has the analytical solution:

$$w_j^* = \frac{1}{\sum_{i=1}^{d} \frac{1}{1+p_i \kappa}} \frac{1}{1+p_j \kappa}, \quad 1 \leq j \leq d. \tag{8}$$

A simple special case occurs when $\sigma_2 = 0$. In this case all $V^t$'s are the same, and the optimal weights go to $w^t = \frac{1}{d}$ for all $t$. In contrast, when $\sigma_2$ grows large the weights eventually converge to $w_d = 1$ and $w^t = 0$ for all $t \neq d$. This results in essentially using only $y^d$ to recover $X^d$ and ignoring the rest of the measurements. Combining these, we note that when the $\sigma_2$ is small, we can gain by a factor of approximately $d$ over the naïve strategy that ignores dynamics and tries to recover $X^d$ using only $y^d$. Notice also that the minimum sample complexity is proportional to $\sum_{t=1}^{d} w_t^2$ when $r/d$ is relatively large. Thus, when $\sigma_2$ is small, the required number of measurements can be reduced by a factor of $d$ compared to what would be required to recover $X^d$ using only $y^d$.

## 3.2 Matrix completion setting

For the matrix completion problem, we consider the following simple uniform sampling ensemble:

**Definition 3.5.** A linear operator $\mathcal{A} : \mathbb{R}^{n_1 \times n_2} \to \mathbb{R}^m$ is a uniform sampling ensemble (with replacement) if all sensing matrices $A_i$ are i.i.d. uniformly distributed on the set

$$\mathcal{X} = \left\{ e_j\left(n_1\right) e_k^T\left(n_2\right), 1 \le j \le n_1, 1 \le k \le n_2 \right),$$

where $e_j\left(n\right)$ are the canonical basis vectors in $\mathbb{R}^n$. We let $p = m_0/(n_1 n_2)$ denote the fraction of sampled entries.

For this observation architecture, our analysis is complicated by the fact that it does not satisfy the matrix RIP. (A quick problematic example is a rank-1 matrix with only one non-zero entry.) To handle this we follow the typical approach and restrict our focus to matrices that satisfy certain *incoherence* properties.

**Definition 3.6.** (Subspace incoherence [10]) Let $U \in \mathbb{R}^{n \times r}$ be the orthonormal basis for an $r$-dimensional subspace $\mathcal{U}$, then the incoherence of $\mathcal{U}$ is defined as $\mu(\mathcal{U}) := \max_{i \in [n]} \frac{\sqrt{n}}{\sqrt{r}} \left\| e_i^T U \right\|_2$, where $e_i$ denotes the $i^{\text{th}}$ standard basis vector. We also simply denote $\mu(\text{span}(U))$ as $\mu(U)$.

**Definition 3.7.** (Matrix incoherence [13]) A rank-$r$ matrix $X \in \mathbb{R}^{n_1 \times n_2}$ with SVD $X = U\Sigma V^T$ is incoherent with parameter $\mu$ if

$$\left\| U_{:i} \right\|_2 \le \frac{\mu\sqrt{r}}{\sqrt{n_1}} \quad \text{for any } i \in [n_1] \quad \text{and} \quad \left\| V_{:j} \right\|_2 \le \frac{\mu\sqrt{r}}{\sqrt{n_2}} \quad \text{for any } j \in [n_2],$$

i.e., the subspaces spanned by the columns of $U$ and $V$ are both $\mu$-incoherent.

The incoherence assumption guarantees that $X$ is far from sparse, which make it possible to recover $X$ from incomplete measurements since a measurement contains roughly the same amount of information for all dimensions.

To proceed we also assume that the matrix $X^d$ has "bounded spikiness" in that the maximum entry of $X^d$ is bounded by $a$, i.e., $\left\| X^d \right\|_\infty \le a$. To exploit the spikiness constraint below we replace the optimization constraints $\mathbb{C}\left(r\right)$ in (3) with $\mathbb{C}\left(r, a\right) := \{ X \in \mathbb{R}^{n_1 \times n_2} : \text{rank}\left(X\right) \le r, \|X\|_\infty \le a \}$:

$$\hat{X}^d = \underset{X \in \mathbb{C}(r,a)}{\arg \min} \mathcal{L}\left(X\right) = \underset{X \in \mathbb{C}(r,a)}{\arg \min} \frac{1}{2} \sum_{t=1}^{d} w_t \left\| \mathcal{A}^t\left(X\right) - y^t \right\|_2^2. \tag{9}$$

Note that Proposition 3.1 still holds for (9).

To obtain an error bound in the matrix completion case, we lower bound the LHS of 4 using a restricted convexity argument (see, for example, [20]) and upper bound the RHS using matrix Bernstein inequality. The result of this approach is the following theorem.

**Theorem 3.8.** *Suppose that we are given measurements as in* (1) *where all $\mathcal{A}^t$'s are uniform sampling ensembles. Assume that $X^t$ evolves according to* (2), *has rank $r$, and is incoherent with parameter $\mu_0$ and $\left\| X^d \right\|_\infty \le a$. Further assume that the perturbation noise and the measurement noise satisfy the same assumptions in Theorem 3.4. If*

$$m_0 \ge D_2 n_{\min} \log^2(n_1 + n_2)\phi'(w), \tag{10}$$

*where $\phi'(w) = \frac{\max_t w_t^2\left((d-t)\mu_0^2 r \sigma_2^2/n_1 + \sigma_1^2\right)}{\sum_{t=1}^{d} w_t^2\left((d-t)\sigma_2^2 + \sigma_1^2\right)}$, then the estimator $\hat{X}^d$ from* (9) *satisfies*

$$\left\| \Delta^d \right\|_F^2 \le \max \left\{ B_1 := C_2 a^2 n_1 n_2 \sqrt{\frac{\sum_{t=1}^{d} w_t^2 \log(n_1 + n_2)}{m_0}}, B_2 \right\}, \tag{11}$$

*with probability at least $P_1 = 1 - 5/(n_1 + n_2) - 5dn_{\max} \exp(-n_{\min})$, where*

$$B_2 = \frac{C_3 r n_1^2 n_2^2 \log(n_1 + n_2)}{n_{\min} m_0} \left( \left( \sum_{t=1}^{d} w_t^2 \sigma_1^2 + \sum_{t=1}^{d-1}(d-t)w_t^2\sigma_2^2 \right) + \sum_{t=1}^{d} w_t^2 a^2 \right), \tag{12}$$

*and $C_2, C_3, D_2$ are absolute positive constants.*

If we choose the weights as $w_d = 1$ and $w_t = 0$ for $1 \leq t \leq d - 1$, the bound in Theorem 3.8 reduces to a result comparable to classical (static) matrix completion results (see, for example, [15] Theorem 7). Moreover, from the $B_2$ term in (11), we obtain the same dependence on $m$ as that of (6), i.e., $1/m$. However, there are also a few key differences between Theorem 3.4 and our results for matrix completion. In general the bound is loose in several aspects compared to the matrix sensing bound. For example, when $m_0$ is small, $B_1$ actually dominates, in which case the dependence on $m$ is actually $1/\sqrt{m}$ instead of $1/m$. When $m_0$ is sufficiently large, then $B_2$ dominates, in which case we can consider two cases. The first case corresponds to when $a$ is relatively large compared to $\sigma_1, \sigma_2$ – i.e., the low-rank matrix is spiky. In this case the term containing $a^2$ in $B_2$ dominates, and the optimal weights are equal weights of $1/d$. This occurs because the term involving $a$ dominates and there is little improvement to be obtained by exploiting temporal dynamics. In the second case, when $a$ is relatively small compared to $\sigma_1, \sigma_2$ (which is usually the case in practice), the bound can be simplified to

$$\|\Delta\|_F^2 \leq \frac{c_3 r n_1^2 n_2^2 \log(n_1 + n_2)}{n_{\min} m_0} \left( \left( \sum_{t=1}^{d} w_t^2 \sigma_1^2 + \sum_{t=1}^{d-1} (d-t) w_t^2 \sigma_2^2 \right) \right).$$

The above bound is much more similar to the bound in (6) from Theorem 3.4. In fact, we can also obtain the optimal weights by solving the same quadratic program as (7).

When $n_1 \approx n_2$, the sample complexity is $\Theta(n_{\min} \log^2(n_1 + n_2) \phi'(w))$. In this case Theorem 3.8 also implies a similar sample complexity reduction as we observed in the matrix sensing setting. However, the precise relations between sample complexity and weights $w_t$'s are different in these two cases (deriving from the fact that the proof uses matrix Bernstein inequalities in the matrix completion setting rather than concentration inequalities of Chi-squared variables as in the matrix sensing setting).

## 4   An algorithm based on alternating minimization

As noted in Section 2, any rank-$r$ matrix can be factorized as $X = UV^T$ where $U$ is $n_1 \times r$ and $V$ is $n_2 \times r$, therefore the LOWEMS estimator in (3) can be reformulated as

$$\hat{X}^d = \arg\min_{X \in \mathcal{C}(r)} \mathcal{L}(X) = \arg\min_{X = UV^T} \sum_{t=1}^{d} \frac{1}{2} w_t \left\| \mathcal{A}^t \left( UV^T \right) - y^t \right\|_2^2. \tag{13}$$

The above program can be solved by alternating minimization (see [17]), which alternatively minimizes the objective function over $U$ (or $V$) while holding $V$ (or $U$) fixed until a stopping criterion is reached. Since the objective function is quadratic, each step in this procedure reduces to conventional weighted least squares, which can be solved via efficient numerical procedures. Theoretical guarantees for global convergence of alternating minimization for the static matrix sensing/completion problem have recently been established in [10, 13, 25] by treating the alternating minimization as a noisy version of the power method. Extending these results to establish convergence guarantees for (13) would involve analyzing a weighted power method. We leave this analysis for future work, but expect that similar convergence guarantees should be possible in this setting.

## 5   Simulations and experiments

### 5.1   Synthetic simulations

Our synthetic simulations consider both matrix sensing and matrix completion, but with an emphasis on matrix completion. We set $n_1 = 100$, $n_2 = 50$, $d = 4$ and $r = 5$. We consider two baselines: **baseline one** is only using $y^d$ to recover $X^d$ and simply ignoring $y^1, \ldots y^{d-1}$; **baseline two** is using $\{y^t\}_{t=1}^{d}$ with equal weights. Note that both of these can be viewed as special cases of LOWEMS with weights $(0, \ldots, 0, 1)$ and $(\frac{1}{d}, \frac{1}{d}, \ldots, \frac{1}{d})$ respectively. Recalling the formula for the optimal choice of weights in (8), it is easy to show that baseline one is equivalent to the case where $\kappa = (\sigma_2^2)/(\sigma_1^2) \to \infty$ and the baseline two equivalent to the case where $\kappa \to 0$. This also makes intuitive sense since $\kappa \to \infty$ means the perturbation is arbitrarily large between time steps, while $\kappa \to 0$ reduces to the static setting.

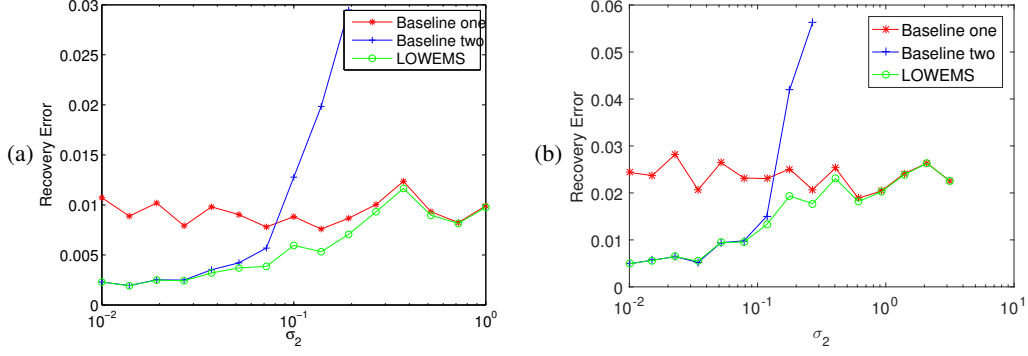

Figure 1: Recovery error under different levels of perturbation noise. (a) matrix sensing. (b) matrix completion.

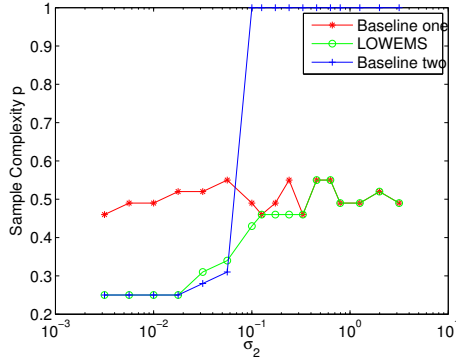

Figure 2: Sample complexity under different levels of perturbation noise (matrix completion).

*1). Recovery error.* In this simulation, we set $m_0 = 4000$ and set the measurement noise level $\sigma_1$ to 0.05. We vary the perturbation noise level $\sigma_2$. For every pair of $(\sigma_1, \sigma_2)$ we perform 10 trials, and show the average relative recovery error $\left\| \Delta^d \right\|_F^2 / \left\| X^d \right\|_F^2$. Figure 1 illustrates how LOWEMS reduces the recovery error compared to our baselines. As one can see, when $\sigma_2$ is small, the optimal $\kappa$, i.e., $\sigma_2^2/\sigma_1^2$, generates nearly equal weights (baseline two), reducing recovery error approximately by a factor of 4 over baseline one, which is roughly equal to $d$ as expected. As $\sigma_2$ grows, the recovery error of baseline two will increase dramatically due to the perturbation noise. However in this case the optimal $\kappa$ of LOWEMS grows with it, leading to a more uneven weighting and to somewhat diminished performance gains. We also note that, as expected, LOWEMS converges to baseline one when $\sigma_2$ is large.

*2). Sample complexity.* In the interest of conciseness we only provide results here for the matrix completion setting (matrix sensing yields broadly similar results). In this simulation we vary the fraction of observed entries $p$ to empirically find the minimum sample complexity required to guarantee successful recovery (defined as a relative error $\leq 0.08$). We compare the sample complexity of the proposed LOWEMS to baseline one and baseline two under different perturbation noise level $\sigma_2$ ($\sigma_1$ is set as 0.02). For a certain $\sigma_2$, the relative recovery error is the averaged over 10 trials. Figure 2 illustrates how LOWEMS reduces the sample complexity required to guarantee successful recovery. When the perturbation noise is weaker than the measurement noise, the sample complexity can be reduced approximately by a factor of $d$ compared to baseline one. When the perturbation noise is much stronger than measurement noise, the recovery error of baseline two will increase due to the perturbation noise and hence the sample complexity increase rapidly. However in this case proposed LOWEMS still achieves relatively small sample complexity and its sample complexity converges to baseline one when $\sigma_2$ is relatively large.

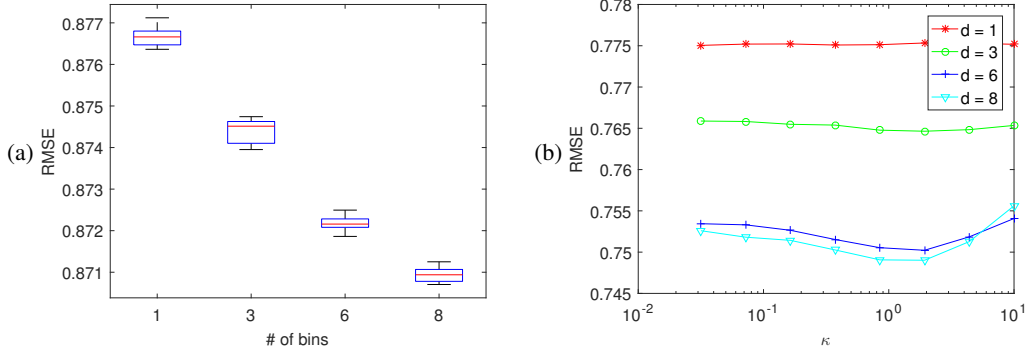

Figure 3: Experimental results on truncated Netflix dataset. (a) Testing RMSE vs. number of time steps. (b) Validation RMSE vs. $\kappa$.

## 5.2 Real world experiments

We next test the LOWEMS approach in the context of a recommendation system using the (truncated) Netflix dataset. We eliminate those movies with few ratings, and those users rating few movies, and generate a truncated dataset with 3199 users, 1042 movies, 2462840 ratings, and hence the fraction of visible entries in the rating matrix is $\approx 0.74$. All the ratings are distributed over a period of 2191 days. For the sake of robustness, we additionally impose a Frobenius norm penalty on the factor matrices $U$ and $V$ in (13). We keep the latest (in time) $10\%$ of the ratings as a testing set. The remaining ratings are split into a validation set and a training set for the purpose of cross validation. We divide the remaining ratings into $d \in \{1, 3, 6, 8\}$ bins respectively with same time period according to their timestamps. We use 5-fold cross validation, and we keep $1/5$ of the ratings from the $d^{\text{th}}$ bin as a validation set. The number of latent factors $r$ is set to 10. The Frobenius norm regularization parameter $\gamma$ is set to 1. We also note that in practice one likely has no prior information on $\sigma_1$, $\sigma_2$ and hence $\kappa$. However, we use model selection techniques like cross validation to select the best $\kappa$ incorporating the unknown prior information on measurement/perturbation noise. We use root mean squared error (RMSE) to measure prediction accuracy. Since alternating minimization uses a random initialization, we generate 10 test RMSE's (using a boxplot) for the same testing set. Figure 3(a) shows that the proposed LOWEMS estimator improves the testing RMSE significantly with appropriate $\kappa$. Additionally, the performance improvement increases as $d$ gets larger.

To further investigate how the parameter $\kappa$ affects accuracy, we also show the validation RMSE compared to $\kappa$ in Figure 3(b). When $\kappa \approx 1$, LOWEMS achieves the best RMSE on the validation data. This further demonstrates that imposing an appropriate dynamic constraint should improve recovery accuracy in practice.

## 6 Conclusion

In this paper we consider the low-rank matrix recovery problem in a novel setting, where one of the factor matrices changes over time. We propose the locally weighted matrix smoothing (LOWEMS) framework, and have established error bounds for LOWEMS in both the matrix sensing and matrix completion cases. Our analysis quantifies how the proposed estimator improves recovery accuracy and reduces sample complexity compared to static recovery methods. Finally, we provide both synthetic and real world experimental results to verify our analysis and demonstrate superior empirical performance when exploiting dynamic constraints in a recommendation system.

### Acknowledge

This work was supported by grants NRL N00173-14-2-C001, AFOSR FA9550-14-1-0342, NSF CCF-1409406, CCF-1350616, and CMMI-1537261.

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
