[Supplementary Material]

# A Proof of Proposition 3.1

*Proof.* Let $x := \text{vec}(X) \in \mathbb{R}^{n_1 n_2}$ and $\tilde{\mathcal{L}}(x) := \mathcal{L}(X)$. Since the objective function is continuous in $X$ and the set $\mathbb{C}(r)$ is compact, $\mathcal{L}(X)$ achieves a minimizer at some point $\hat{X}^d \in \mathbb{C}(r)$.

Since $\hat{X}^d$ is a minimizer of the constrained problem, then for all matrices $X \in \mathbb{C}(r)$ we have the following inequality

$$\tilde{\mathcal{L}}(\hat{x}^d) - \tilde{\mathcal{L}}(x) \leq 0. \tag{14}$$

By the second-order Taylor's theorem, we expand $\tilde{\mathcal{L}}(x)$ around $x^d = \text{vec}(X^d)$

$$\tilde{\mathcal{L}}(x) = \tilde{\mathcal{L}}(x^d) + \left\langle \nabla \tilde{\mathcal{L}}(x^d), x - x^d \right\rangle + \frac{1}{2} \left\langle \nabla^2 \tilde{\mathcal{L}}(\bar{x})(x - x^d), x - x^d \right\rangle, \tag{15}$$

where $\bar{x} = \alpha x^d + (1 - \alpha) x$ for some $\alpha \in [0, 1]$. Plugging (15) with $x = \hat{x}^d$ into (14) we obtain

$$\left\langle \nabla \tilde{\mathcal{L}}(x^d), \hat{x}^d - x^d \right\rangle + \frac{1}{2} \left\langle \nabla^2 \tilde{\mathcal{L}}(\bar{x})(\hat{x}^d - x^d), \hat{x}^d - x^d \right\rangle \leq 0. \tag{16}$$

Through some algebraic manipulation we have the following expression for the gradient of $\tilde{\mathcal{L}}(x)$:

$$\nabla \tilde{\mathcal{L}}(x) = \text{vec}\left( \sum_{t=1}^{d} w_t \mathcal{A}^{t*} \left[ \mathcal{A}^t(X) - y^t \right] \right). \tag{17}$$

Based on the above gradient it follows that

$$\nabla^2 \tilde{\mathcal{L}}(x) b = \text{vec}\left( \sum_{t=1}^{d} w_t \mathcal{A}^{t*} \left[ \mathcal{A}^t(B) \right] \right), \tag{18}$$

where $b = \text{vec}(B)$.

Now based on (17) and (18), the absolute value of first term in (16) can be bounded as

$$\left| \left\langle \nabla \tilde{\mathcal{L}}(x^d), \hat{x}^d - x^d \right\rangle \right| = \left| \left\langle \sum_{t=1}^{d} w_t \mathcal{A}^{t*} \left[ \mathcal{A}^t(X^d) - y^t \right], \Delta^d \right\rangle \right|$$

$$\leq \left\| \sum_{t=1}^{d} w_t \mathcal{A}^{t*} \left[ \mathcal{A}^t(X^d) - y^t \right] \right\|_2 \left\| \Delta^d \right\|_* \tag{19}$$

$$\leq \left\| \sum_{t=1}^{d} w_t \mathcal{A}^{t*} \left( h^t - z^t \right) \right\|_2 \sqrt{2r} \left\| \Delta^d \right\|_F$$

The first inequality above used the trace dual norm inequality, while the second inequality follows from a basic inequality for rank-$2r$ matrices. Similarly the second term in (16) is

$$\frac{1}{2} \left\langle \nabla^2 \tilde{\mathcal{L}}(\bar{x})(\hat{x}^d - x^d), \hat{x}^d - x^d \right\rangle = \frac{1}{2} \left\langle \sum_{t=1}^{d} w_t \mathcal{A}^{t*} \mathcal{A}^t(\Delta^d), \Delta^d \right\rangle$$

$$= \frac{1}{2} \sum_{t=1}^{d} w_t \left\langle \mathcal{A}^t(\Delta^d), \mathcal{A}^t(\Delta^d) \right\rangle. \tag{20}$$

The result follows from combining (19) and (20). Note that the above proof holds if we replace $\mathbb{C}(r,)$ with $\mathbb{C}(r, a)$, which completes our proof. $\square$

# B Proof of Theorem 3.4

*Proof.* The proof consists of lower bounding the LHS of (4) and upper bounding the RHS of (4).

We use the following lemma to lower bound $\sum_{t=1}^{d} w_t \left\| \mathcal{A}^t(\Delta^d) \right\|_2^2$.

**Lemma B.1.** *Suppose the linear operator $\mathcal{A}^t : \mathbb{R}^{n_1 \times n_2} \to \mathbb{R}^{m_0}$ is random Gaussian ensemble for all $1 \le t \le d$. If $m_0 > Dn_{\max} r \sum_{t=1}^d w_t^2$, the composite operator $\left\{ \sqrt{w_t} \mathcal{A}^t \right\}_{t=1}^d$ satisfies the rank-2r matrix RIP with constant $\delta_{2r} \le \delta$ with probability exceeding $1 - C \exp\left(-cm_0\right)$, where $D, C$ and $c$ (which depends on $\sigma$) are absolute positive constants.*

*Proof.* See Appendix C. □

Next lemma gives us an upper bound for the stochastic error $\left\| \sum_{t=1}^d w_t \mathcal{A}^{t*} \left( h^t - z^t \right) \right\|_2$.

**Lemma B.2.** *Under the assumptions of Theorem 3.4, when $m_0 \ge Dn_{\max}$, we have*

$$\left\| \sum_{t=1}^d w_t \mathcal{A}^{t*} \left( h^t - z^t \right) \right\|_2 \le C_1 \sqrt{n_{\max}(1 + \delta_1) \left( \sum_{t=1}^d w_t^2 \sigma_1^2 + \sum_{t=1}^{d-1} (d-t) w_t^2 \frac{2rn_2}{m_0} \sigma_2^2 \right)}$$

*with probability exceeding $1 - dC \exp(-cn_2)$, where $D, C_1, C, c$ are positive constants and $\delta_1$ is the rank-1 matrix RIP parameter for all $\mathcal{A}^t$'s.*

*Proof.* See Appendix D. □

Theorem 3.4 follows by combining Lemma B.1, Lemma B.2 and Definition 3.3. □

## C Proof of Lemma B.1

*Proof.* First we introduce the following theorem providing a double-sided tail bound on the sum of independent sub-exponential random variables.

**Theorem C.1.** *For independent $X_i$ sub-exponential with parameters $(\sigma_i, b_i)$, with mean $\mu_i$,*

$$\mathbb{P}\left( \left| \sum_{i=1}^n (X_i - \mu_i) \right| \ge nt \right) \le 2 \exp\left( -\frac{nt^2}{2(\sigma^2 + bt)} \right),$$

*where $\sigma^2 = \sum_i \sigma_i^2$ and $b = \max_i b_i$.*

We now lower bound $\sum_{t=1}^d w_t \left\| \mathcal{A}^t \left( \Delta^d \right) \right\|_2^2$. Since all $\mathcal{A}^t$'s are Gaussian random measurement ensembles, then a particular measurement $\left\langle A_i^t, \Delta^d \right\rangle^2$ is distributed as $m_0^{-1} \left\| \Delta^d \right\|_F^2 \chi^2(1)$. Therefore $\sum_{t=1}^d w_t \left\| \mathcal{A}^t \left( \Delta^d \right) \right\|_2^2 = \sum_{t,i} w_t \left\langle A_i^t, \left( \Delta^d \right) \right\rangle^2$ is a weighted sum of i.i.d. $\chi^2(1)$ random variables. Since $\chi^2(1)$ is sub-exponential with parameters $(4, 4)$, Theorem C.1 implies a double-sided tail bound for $\sum_{t=1}^d w_t \left\| \mathcal{A}^t \left( \Delta^d \right) \right\|_2^2$: for any given $\Delta^d \in \mathbb{R}^{n_1 \times n_2}$ and any fixed $0 < s < 1$

$$\mathbb{P}\left( \left| \sum_{t=1}^d w_t \left\| \mathcal{A}^t \left( \Delta^d \right) \right\|_2^2 - \left\| \Delta^d \right\|_F^2 \right| \le s \left\| \Delta^d \right\|_F^2 \right) \le 2 \exp\left( -\frac{m_0 s^2}{8 \sum_{t=1}^d w_t^2 + 8w_{\max} s} \right),$$

where $w_{\max} = \max\{w_1, \ldots, w_d\}$. The probability can be further simplified if $s$ is very small $(\le 1/d)$.

Rank of $\Delta^d$ is at most $2r$ since $\hat{X}^d, X^d$ are rank-$r$ matrices. By Theorem 2.3 in [4] (one may see the proof if necessary) if $m_0 > Dn_{\max} r \sum_{t=1}^d w_t^2$, the composite operator $\left\{ \sqrt{w_t} \mathcal{A}^t \right\}_{t=1}^d$ satisfies the rank-2r matrix RIP with constant $\delta_{2r} \le \delta$ with probability exceeding $1 - C \exp\left(-cm_0\right)$, where $C$ and $c$ (depends on $\delta$) are absolute positive constants. □

## D  Proof of Lemma B.2

*Proof.* Let $W = \sum_{t=1}^{d} w_t \mathcal{A}^{t*} \left( h^t - z^t \right)$ and $n = n_{\max}$ for short. Following the basic framework of the proof of Lemma 1.1 in [4], we use $\epsilon$-nets method to bound the stochastic error $\|W\|_2$. The operator norm of $W$ is

$$\|W\|_2 = \sup_{\|u\|=\|v\|=1} \langle u, Wv \rangle,$$

Consider a $1/4$-net $\mathcal{N}_{1/4}$ of the unite sphere $S^{n-1}$ with $\left| \mathcal{N}_{1/4} \right| \le 12^n$ (see (III.1) in [4]). For any $v, u \in S^{n-1}$

$$\begin{aligned}
\langle u, Wv \rangle &= \langle u - u_0, Wv \rangle + \langle u_0, W(v - v_0) \rangle + \langle u_0, Wv_0 \rangle \\
&\le \|W\|_2 \|u - u_0\|_2 + \|W\|_2 \|v - v_0\|_2 + \langle u_0, Wv_0 \rangle,
\end{aligned}$$

for some $v_0, w_0 \in \mathcal{N}_{1/4}$ obeying $\|u - u_0\|_2 \le 1/4$ and $\|v - v_0\| \le 1/4$. So the operator norm of $W$ is

$$\|W\|_2 \le 2 \sup_{u_0, v_0 \in \mathcal{N}_{1/4}} \langle u_0, Wv_0 \rangle.$$

For fixed $u_0, v_0$

$$\langle u_0, Wv_0 \rangle = \mathrm{Tr}\left( u_0^T Wv_0 \right) = \mathrm{Tr}\left( v_0 u_0^T W \right) = \left\langle u_0 v_0^T, W \right\rangle = \sum_{t=1}^{d} w_t \left\langle \mathcal{A}^t \left( u_0 v_0^T \right), h^t - z^t \right\rangle.$$

Let $Z = \sum_{t=1}^{d} w_t \left\langle \mathcal{A}^t \left( u_0 v_0^T \right), z^t \right\rangle$ and $H = \sum_{t=1}^{d} w_t \left\langle \mathcal{A}^t \left( u_0 v_0^T \right), h^t \right\rangle$. Since for all $1 \le t \le d$, entries of $z^t$ are i.i.d. $\mathcal{N}\left( 0, \sigma_1^2 \right)$, therefore $Z \sim \mathcal{N}\left( 0, \sigma_Z^2 \right)$, where the variance $\sigma_Z^2$ is

$$\sigma_Z^2 = \sum_{t=1}^{d} w_t^2 \left\| \mathcal{A}^t \left( u_0 v_0^T \right) \right\|_2^2 \sigma_1^2 \le \sum_{t=1}^{d} w_t^2 \left( 1 + \delta_1 \right) \left\| u_0 v_0^T \right\|_F^2 \sigma_1^2 = \sum_{t=1}^{d} w_t^2 \left( 1 + \delta_1 \right) \sigma_1^2. \quad (21)$$

The first inequality uses the matrix RIP for rank-1 matrices. For a fixed $t$, $\mathcal{A}^t$ satisfies the rank-1 matrix RIP with constant $\delta_1$, with probability at least $1 - C_2 \exp(-c_2 m_0)$ provided that $m_0 \ge D_2 n$ by Theorem 2.3 in [4], where $C_2, c_2$ and $D_2$ are fixed positive constants. Then by a union bound, for all $1 \le t \le d$, $\mathcal{A}^t$ satisfies the rank-1 matrix RIP property with parameter $\sigma_1$, with probability at least $1 - dC_2 \exp(-c_2 m_0)$ provided that $m_0 \ge D_2 n$.

We now simplify $H$ as

$$\begin{aligned}
H = \sum_{t=1}^{d} w_t \left\langle \mathcal{A}^t \left( u_0 v_0^T \right), h^t \right\rangle &= \sum_{t=1}^{d-1} w_t \left\langle \mathcal{A}^t \left( u_0 v_0^T \right), \sum_{s=t+1}^{d} \mathcal{A}^t \left[ U \left( \epsilon^s \right)^T \right] \right\rangle \\
&= \sum_{s=2}^{d} \sum_{t=1}^{s-1} \left\langle w_t \mathcal{A}^t \left( u_0 v_0^T \right), \mathcal{A}^t \left[ U \left( \epsilon^s \right)^T \right] \right\rangle \\
&= \sum_{s=2}^{d} \sum_{t=1}^{s-1} \left\langle w_t \mathcal{A}^{t*} \mathcal{A}^t \left( u_0 v_0^T \right), U \left( \epsilon^s \right)^T \right\rangle \\
&= \sum_{s=2}^{d} \sum_{t=1}^{s-1} \sum_{i=1}^{m_0} \left\langle w_t \left[ \mathcal{A}^t \left( u_0 v_0^T \right) \right]_i A_i^t, U \left( \epsilon^s \right)^T \right\rangle \\
&= \sum_{s=2}^{d} \left\langle \sum_{t=1}^{s-1} w_t \left\| \mathcal{A}^t \left( u_0 v_0^T \right) \right\|_2 U^T A^t, \left( \epsilon^s \right)^T \right\rangle,
\end{aligned}$$

where $A^t \in \mathbb{R}^{n_1 \times n_2}$ contains i.i.d. $\mathcal{N}\left( 0, 1/m_0 \right)$ entries. The last equality uses the property that sum of independent Gaussian variables is also Gaussian, and the variance is the sum of individual variances. Since for all $2 \le s \le d$, entries of $\epsilon^s$ are i.i.d. $\mathcal{N}\left( 0, \sigma_2^2 \right)$, therefore $H \sim \mathcal{N}\left( 0, \sigma_H^2 \right)$,

where the variance $\sigma_H^2$ is

$$\sigma_H^2 = \sum_{s=2}^{d}\left\|\sum_{t=1}^{s-1}w_t\left\|\mathcal{A}^t\left(u_0v_0^T\right)\right\|_2 U^T A^t\right\|_F^2 \sigma_2^2 \overset{(\xi_1)}{\leq} \sum_{s=2}^{d}\left\|\sum_{t=1}^{s-1}w_t\sqrt{1+\delta_1}U^T A^t\right\|_F^2 \sigma_2^2$$

$$\overset{(\xi_2)}{=} \sum_{s=2}^{d}\sum_{t=1}^{s-1}w_t^2\left(1+\delta_1\right)\left\|U^T B^s\right\|_F^2 \sigma_2^2$$

$$= \sum_{s=2}^{d}\sum_{t=1}^{s-1}w_t^2\left(1+\delta_1\right)\frac{1}{m_0}\chi_s^2\left(rn_2\right)\sigma_2^2 \qquad (22)$$

$$\overset{(\xi_3)}{\leq} \sum_{s=2}^{d}\sum_{t=1}^{s-1}w_t^2\left(1+\delta_1\right)\frac{1}{m_0}3m_0\sigma_2^2$$

$$= \sum_{t=1}^{d-1}(d-t)w_t^2\left(1+\delta_1\right)\sigma_2^2.$$

Inequality $(\xi_1)$ holds with probability exceeding $1-dC_2\exp(-c_2m_0)$ provided that $m_0 \geq Dn$ based on the matrix RIP for rank-1 matrices as used while bounding $\sigma_Z^2$. Equality $(\xi_2)$ uses the property that sum of independent Gaussian variables is also Gaussian and entries of $B^s$ are i.i.d. $\mathcal{N}(0,1/m_0)$. Inequality $(\xi_3)$ holds with probability at least $1-dC_3\exp(-c_3m_0)$ by the concentration property of correlated Chi-squared variables.

Since the measurement noise $Z$ and dynamic perturbation $H$ are independent, then $\langle u_0, Wv_0\rangle \sim \mathcal{N}\left(0, \sigma_Z^2 + \sigma_H^2\right)$. Then by a standard tail bound for Gaussian random variables we have

$$\mathbb{P}\left(|\langle u_0, Wv_0\rangle| > \lambda\right) \leq 2\exp\left(-\frac{\lambda^2}{2\left(\sigma_H^2 + \sigma_Z^2\right)}\right).$$

Therefore by an standard union bound we bound the stochastic error

$$\mathbb{P}\left(\|W\|_2 \geq C_0\sqrt{n\left(\sigma_H^2 + \sigma_Z^2\right)}\right) \leq 2\left|\mathcal{N}_{1/4}\right|^2\exp\left(-\frac{C_0^2n}{8}\right) \leq 2\exp\left(-cn\right), \qquad (23)$$

where $c = \frac{C_0^2}{8} - 2\log 12$. To ensure $c > 0$, we require $C_0 > 4\sqrt{\log 12}$.

Combining (21), (22), and (23), if $m_0 \geq Dn$ we have

$$\|W\|_2 \leq C_0\sqrt{n\left((1+\delta_1)\sum_{t=1}^{d}w_t^2\left(\sigma_1^2 + (d-t)\frac{5rn_2}{m_0}\sigma_2^2\right)\right)}$$

with probability exceeding $1 - [dC_2\exp(-c_2m_0) + dC_3\exp(-c_3m_0) + 2\exp(-cn)] \geq 1 - dC\exp(-cn_2)$. $\qquad\square$

## E  Proof of Theorem 3.8

*Proof.* The proof follows the same framework of the proof of Theorem 7 in [15].

Before we lower bound $\sum_{t=1}^{d}w_t\left\|\mathcal{A}^t\left(\Delta^d\right)\right\|_2^2$, we consider the following constraint set for a given $0 < r \leq n$:

$$\mathcal{E}(r) = \left\{X \in \mathbb{C}(r) : \|X\|_\infty = 1, \|X\|_F^2 \geq n_1n_2\sqrt{\frac{2048\sum_{t=1}^{d}w_t^2\log(n_1+n_2)}{\log(6/5)m_0}}\right\}.$$

Define the following random matrix

$$\Sigma_R = \sum_{t=1}^{d}\sum_{i=1}^{m_0}w_t\gamma_i^t A_i^t,$$

where $\gamma_i^t$ is Rademacher variable.

The following lemma bounds the restricted strong convexity (see [20]) of the operator $\left\{\sqrt{w_t}\mathcal{A}^t\right\}_{t=1}^{d}$.

**Lemma E.1.** *Suppose all $\mathcal{A}^t$'s are fixed uniform sampling ensembles. For all $X \in \mathcal{E}(r)$*

$$\sum_{t=1}^{d} w_t \left\| \mathcal{A}^t(X) \right\|_2^2 \geq \frac{p}{2} \|X\|_F^2 - \frac{44 r n_1 n_2}{m_0} \left( \mathbb{E}(\|\Sigma_R\|) \right)^2 \tag{24}$$

*with probability at least $1 - \frac{2}{(n_1+n_2)}$.*

*Proof.* See Appendix F. □

Note that $\|\Delta^d\|_\infty \leq \|\hat{X}^d\|_\infty + \|X^d\|_\infty \leq 2 \|X^d\|_\infty$. To proceed, we consider the following two cases.

*Case I.* $\frac{\Delta^d}{2\|X^d\|_\infty} \notin \mathcal{E}(2r)$.

Following the definition of $\mathcal{E}(2r)$ we have

$$\left\| \Delta^d \right\|_F^2 \leq c_2 \left\| X^d \right\|_\infty^2 n_1 n_2 \sqrt{\frac{\sum_{t=1}^{d} w_t^2 \log(n_1+n_2)}{m_0}},$$

where $C_2 = 4 \sqrt{\frac{2048}{\log(6/5)}}$. This yields the first part of inequality (11) in Theorem 3.8.

*Case II.* $\frac{\Delta^d}{2\|X^d\|_\infty} \in \mathcal{E}(2r)$.

Since $\frac{\Delta^d}{2\|X^d\|_\infty} \in \mathcal{E}(2r)$, applying Lemma E.1 yields

$$\sum_{t=1}^{d} w_t \left\| \mathcal{A}^t(\Delta^d) \right\|_2^2 \geq \frac{p}{2} \left\| \Delta^d \right\|_F^2 - \frac{362 r n_1 n_2}{m_0} \left( \mathbb{E}(\|\Sigma_R\|) \right)^2 \left\| X^d \right\|_\infty^2. \tag{25}$$

Combining (25) and (4) yields

$$\frac{p}{2} \left\| \Delta^d \right\|_F^2 \leq 2\sqrt{2r} \left\| \sum_{t=1}^{d} w_t \mathcal{A}^{t*}(h^t - z^t) \right\|_2 \left\| \Delta^d \right\|_F + \frac{362 r n_1 n_2}{m_0} \left( \mathbb{E}(\|\Sigma_R\|) \right)^2 \left\| X^d \right\|_\infty^2$$

$$\leq \frac{8r}{p} \left\| \sum_{t=1}^{d} w_t \mathcal{A}^{t*}(h^t - z^t) \right\|_2^2 + \frac{p}{4} \left\| \Delta^d \right\|_F^2 + \frac{362 r n_1 n_2}{m_0} \left( \mathbb{E}(\|\Sigma_R\|) \right)^2 \left\| X^d \right\|_\infty^2.$$

The above inequality can be further simplified as

$$\left\| \Delta^d \right\|_F^2 \leq \frac{32 r n_1^2 n_2^2}{m_0^2} \left\| \sum_{t=1}^{d} w_t \mathcal{A}^{t*}(h^t - z^t) \right\|_2^2 + \frac{1448 r n_1^2 n_2^2}{m_0^2} \left( \mathbb{E}(\|\Sigma_R\|) \right)^2 \left\| X^d \right\|_\infty^2. \tag{26}$$

Next we bound $\mathbb{E}(\|\Sigma_R\|)$ in the following lemma.

**Lemma E.2.** *Suppose all $\mathcal{A}^t$'s are fixed uniform sampling ensembles. For $m_0 \geq D n_{\min} \log(n_1+n_2) \phi(w)$, where $\phi(w) = \frac{w_{\max}^2}{\sum_{t=1}^{d} w_t^2}$, there exists an absolute positive constant $C$ such that*

$$\mathbb{E}(\|\Sigma_R\|) \leq C \sqrt{\frac{2e \log(n_1+n_2) \sum_{t=1}^{d} w_t^2 m_0}{n_{\min}}}. \tag{27}$$

The proof is not provided since it is almost the same as that of Lemma 6 in [15] with some minor modifications. Note that our results are a bit stronger compared to Lemma 6 in [15], since we are dealing with bounded variables.

Now we upper bound the stochastic error $\|J\|_2^2 := \left\| \sum_{t=1}^{d} w_t \mathcal{A}^{t*}(h^t - z^t) \right\|_2^2$. First, we rewrite $J$ as

$$J = \sum_{t=1}^{d} w_t \mathcal{A}^{t*} \mathcal{A}^t \left[ U \left( \sum_{s=t+1}^{d} \epsilon^s \right)^T + Z^t \right],$$

where each entry of the random matrix $Z^t \in \mathbb{R}^{n_1 \times n_2}$ is i.i.d. Gaussian distributed with variance $\sigma_1^2$. Set $Y^t = U \left( \sum_{s=t+1}^d \epsilon^s \right)^T$ and $F^t = Y^t + Z^t$. Note that $F^t$ may be correlated for different $1 \le t \le d$, though for a given $t$ the entries of $F^t$ are independent.

We now introduce an $n_1 \times n_2$ random matrix $G^t$ that has exactly one non-zero entry:

$$G^t = w_t n_1 n_2 F_{ij}^t E_{ij}, \quad \text{with probability } \frac{1}{n_1 n_2},$$

where $E_{ij}$ is the canonical basis of matrices with dimension $n_1 \times n_2$. We also introduce the following random matrix $H^t$, which is the average of $m_0$ independent copies of $G^t$:

$$H^t = \frac{1}{m_0} \sum_{i=1}^{m_0} G_i^t \quad \text{where each } G_i^t \text{ is an independent copy of } G^t.$$

Then $J$ can be decomposed as sum of independent random matrices: $J = \frac{m_0}{n_1 n_2} \sum_{t=1}^d H^t$. It is immediate that

$$\mathbb{E}G^t = \mathbb{E}H^t = w_t F^t, \quad \mathbb{E}J = \frac{m_0}{n_1 n_2} \sum_{t=1}^d w_t F^t.$$

Before we proceed we introduce a lemma describing the spectral norm deviation of a sum of uncentered random matrices from its mean value.

**Lemma E.3.** *(Corollary 6.1.2 in [24]) Consider a finite sequence $\{S_k\}$ of independent random matrices with common dimension $n_1 \times n_2$. Assume that each matrix has uniformly bounded deviation from its mean:*
$$\|S_k - \mathbb{E}S_k\| \le L \quad \text{for each index } k.$$

*Consider the sum*
$$Z = \sum_k S_k.$$

*Let $\rho(Z)$ denotes the matrix variance statistic of the sum:*
$$\rho(Z) = \max \left\{ \left\| \mathbb{E}[(Z - \mathbb{E}Z)(Z - \mathbb{E}Z)^T] \right\|, \left\| \mathbb{E}[(Z - \mathbb{E}Z)^T(Z - \mathbb{E}Z)] \right\| \right\}$$
$$= \max \left\{ \left\| \sum_k \mathbb{E}[(S_k - \mathbb{E}S_k)(S_k - \mathbb{E}S_k)^T] \right\|, \left\| \sum_k \mathbb{E}[(S_k - \mathbb{E}S_k)^T(S_k - \mathbb{E}S_k)] \right\| \right\}.$$

*Then for all $s \ge 0$,*
$$\mathbb{P}(\|Z - \mathbb{E}Z\| \ge s) \le (n_1 + n_2) \exp\left( \frac{-s^2/2}{\rho(Z) + Ls/3} \right).$$

We are going to apply the above uncentered Bernstein inequality to the sum of $dm_0$ independent random matrices $\sum_{t=1}^d H^t = \frac{1}{m_0} \sum_{t=1}^d \sum_{k=1}^{m_0} G_k^t$. Before doing so, we note that for given $t$ and $k$,

$$\left\| G_k^t - \mathbb{E}G_k^t \right\| \le \left\| G_k^t \right\| + \left\| \mathbb{E}G_k^t \right\| \le \left\| G_k^t \right\| + \mathbb{E}\left\| G_k^t \right\| \le 2 \left\| G_k^t \right\|.$$

The first inequality uses the triangle inequality; the second is Jensen's inequality.

To control $\rho(\sum_{t=1}^d H^t)$, first note that

$$\mathbf{0} \preceq \sum_t \sum_k \mathbb{E}\left[ G_k^t - \mathbb{E}G_k^t)(G_k^t - \mathbb{E}G_k^t)^T \right] = \sum_t \sum_k \mathbb{E}\left[ (G_k^t (G_k^t)^T) - (\mathbb{E}G_k^t)(\mathbb{E}G_k^t)^T \right]$$
$$\preceq \sum_t \sum_k \mathbb{E}\left[ G_k^t (G_k^t)^T \right]$$
$$= m_0 \sum_t \mathbb{E}\left[ G^t (G^t)^T \right].$$

The third relation holds because $(\mathbb{E}G_k^t)(\mathbb{E}G_k^t)^T$ is positive semidefinite; the last relation uses the fact that for a fixed $t$, $G_k^t$ are random matrices following identical distributions independently for all $1 \le k \le m_0$. Now we can control $\rho(\sum_{t=1}^d H^t)$ in the following

$$\rho\left(\sum_{t=1}^d H^t\right) \le \frac{1}{m_0} \max\left\{\left\|\sum_t \mathbb{E}\left[(G^t(G^t)^T)\right]\right\|, \left\|\sum_t \mathbb{E}\left[(G^t)^T G^t\right]\right\|\right\}.$$

Set $\rho_0 := \max\left\{\left\|\sum_{t=1}^d \mathbb{E}(G^t(G^t)^T)\right\|, \left\|\sum_{t=1}^d \mathbb{E}((G^t)^T G^t)\right\|\right\}$. Then the remaining work is to uniformly upper bound $\|G_k^t\|$ for all $1 \le t \le d$ and $1 \le k \le m_0$ and upper bound $\rho_0$.

First we turn to the uniform bound on the spectral norm of the random matrix $G_k^t$ for all $1 \le t \le d$ and $1 \le k \le m_0$. We have for all $1 \le t \le d$ and $1 \le k \le m_0$

$$\left\|G_k^t\right\| \le \max_{i,j,t} w_t \left\|n_1 n_2 F_{ij}^t E_{ij}\right\| = n_1 n_2 \max_{i,j,t} w_t |F_{ij}^t|.$$

Since $\mu(U) \le \mu_0$, the variance of each entry of the random matrix $F^t$ can be bounded as $\mathrm{Var}(F_{ij}^t) \le \frac{\mu_0^2 r}{n_1} \sigma_2^2 (d-t) + \sigma_1^2$. Let $\sigma_{\max}^2 = \max_t w_t^2 \left(\frac{\mu_0^2 r}{n_1}\sigma_2^2(d-t) + \sigma_1^2\right)$. Then by the tail probability of Gaussian random variables and the standard union bound (over $i,j$), for all $1 \le t \le d$ and $1 \le k \le m_0$ we have

$$\mathbb{P}\left(\left\|G_k^t\right\| \le n_1 n_2 \sqrt{2\log(d(n_1+n_2)n_1 n_2)\sigma_{\max}^2} =: L\right) \ge 1 - 2/(n_1 + n_2).$$

Second we turn to the computation of $\mathbb{E}(G^t(G^t)^T)$. We have

$$\mathbb{E}(G^t(G^t)^T) = w_t^2 n_1^2 n_2^2 \sum_{i=1}^{n_1}\sum_{j=1}^{n_2}(F_{ij}^t)^2 E_{ij}E_{ij}^T \frac{1}{n_1 n_2} = w_t^2 n_1 n_2 \sum_{i=1}^{n_1}\sum_{j=1}^{n_2}(F_{ij}^t)^2 E_{ii}.$$

Similarly $\mathbb{E}((G^t)^T G^t) = w_t^2 n_1 n_2 \sum_{i=1}^{n_1}\sum_{j=1}^{n_2}(F_{ij}^t)^2 E_{jj}$. Then

$$\rho = n_1 n_2 \max\left\{\max_i \sum_{t=1}^d \sum_{j=1}^{n_2} w_t^2 (F_{ij}^t)^2, \max_j \sum_{t=1}^d \sum_{i=1}^{n_1} w_t^2 (F_{ij}^t)^2\right\}.$$

Let $a_i = \sum_{t=1}^d \sum_{j=1}^{n_2} w_t^2 (F_{ij}^t)^2$ and $b_j = \sum_{t=1}^d \sum_{i=1}^{n_1} w_t^2 (F_{ij}^t)^2$. We first bound $\max_i a_i$. Note that $a_i = \sum_{t=1}^d w_t^2 \sum_{j=1}^{n_2}(Y_{ij}^t + Z_{ij}^t)^2 \le 2\sum_{t=1}^d w_t^2 \sum_{j=1}^{n_2}[(Y_{ij}^t)^2 + (Z_{ij}^t)^2]$. Note that for $1 \le i \le n_1$ and $1 \le t \le d$, $\sum_{j=1}^{n_2}(Z_{ij}^t)^2 \sim \sigma_1^2 \chi^2(n_2)$ and are independent. So by the tail bound of Chi-squared variable and the standard union bound (over $i$ and $t$) we have

$$\mathbb{P}\left(\max_i \sum_{t=1}^d w_t^2 \sum_{j=1}^{n_2}(Z_{ij}^t)^2 \le 5n_2 \sum_{t=1}^d w_t^2 \sigma_1^2\right) \ge 1 - dn_1 \exp(-n_2). \tag{28}$$

Similarly we have

$$\mathbb{P}\left(\max_j \sum_{t=1}^d w_t^2 \sum_{i=1}^{n_2}(Z_{ij}^t)^2 \le 5n_1 \sum_{t=1}^d w_t^2 \sigma_1^2\right) \ge 1 - dn_2 \exp(-n_1). \tag{29}$$

For $\sum_{j=1}^{n_2}(Y_{ij}^t)^2$, note that $Y_{ij}^t$ is Gaussian distributed and the variance is not greater than $\frac{\mu_0^2 r}{n_1}(d-t)\sigma_2^2$ for all $i,j,t$, since $\mu(U) \le \mu_0$. For a fixed $i$, for all $1 \le j \le n_2$, $Y_{ij}^t$ are independent Gaussian random variables. So given $i$ and $t$, applying the tail bound of Chi-squared random variables yields

$$\mathbb{P}\left(\sum_{j=1}^{n_2}(Y_{ij}^t)^2 \le 5n_2(d-t)\frac{\mu_0^2 r}{n_1}\sigma_2^2\right) \ge 1 - \exp(-n_2).$$

By the standard union bound (over $i$ and $t$) we have

$$\mathbb{P}\left(\max_i \sum_{t=1}^d w_t^2 \sum_{j=1}^{n_2}(Y_{ij}^t)^2 \le 5n_2 \frac{\mu_0^2 r}{n_1}\sum_{t=1}^d (d-t)w_t^2\sigma_2^2\right) \ge 1 - dn_1 \exp(-n_2). \tag{30}$$

Now we turn to $\sum_{i=1}^{n_1}(Y_{ij}^t)^2$, which follows a Chi-squared distribution $(d-t)\sigma_2^2\chi^2(r)$, since

$$\sum_{i=1}^{n_1}(Y_{ij}^t)^2 = (Y_{:j}^t)^T Y_{:j}^t = \bar\epsilon_{j:}^t U^T U \left(\bar\epsilon_{j:}^t\right)^T = \bar\epsilon_{j:}^t \left(\bar\epsilon_{j:}^t\right)^T$$

where $\bar\epsilon^t = \sum_{s=t+1}^d \epsilon^s$. The last equality uses the fact that $U$ is orthonormal. Then by the tail bound of Chi-squared random variables and the standard union bound (over $j$ and $t$) we have

$$\mathbb{P}\left(\max_j \sum_{t=1}^d w_t^2 \sum_{i=1}^{n_1}(Y_{ij}^t)^2 \le 5n_1 \sum_{t=1}^d (d-t)w_t^2\sigma_2^2\right) \ge 1 - dn_2\exp(-n_1). \tag{31}$$

Combining (28) and (30) yields

$$\mathbb{P}\left(\max_i a_i \le 10n_2\sum_{t=1}^d w_t^2\left(\sigma_1^2 + \frac{\mu_0^2 r}{n_1}(d-t)\sigma_2^2\right)\right) \ge 1 - 2dn_1\exp(-n_2). \tag{32}$$

Similarly combining (29) and (31) yields

$$\mathbb{P}\left(\max_j b_j \le 10n_1\sum_{t=1}^d w_t^2\left(\sigma_1^2 + (d-t)\sigma_2^2\right)\right) \ge 1 - 2dn_2\exp(-n_1). \tag{33}$$

Note that $1 \le \mu_0 \le \sqrt{n_1}/\sqrt{r}$, so $\frac{\mu_0^2 r}{n_1} \le 1$. Now we are ready to bound $\rho_0$ by combining (32) and (33):

$$\mathbb{P}\left(\rho_0 \le 10n_{\max}n_1 n_2\left(\sum_{t=1}^d w_t^2\sigma_1^2 + \sum_{t=1}^d w_t^2(d-t)\sigma_2^2\right) =: \nu\right) \ge 1 - 4dn_{\max}\exp(-n_{\min}). \tag{34}$$

Now by Lemma E.3, we have

$$\mathbb{P}\left(\left\|\sum_{t=1}^d H^t - \sum_{t=1}^d w_t F^t\right\| \ge s\right) \le (n_1+n_2)\exp\left(\frac{-m_0 s^2/2}{\nu + 2Ls/3}\right).$$

If we let $s = \sqrt{\frac{8\log(n_1+n_2)\nu}{m_0}}$ and substitute this into the above matrix Bernstein inequality we obtain

$$\mathbb{P}\left(\left\|\sum_{t=1}^d H^t - \sum_{t=1}^d w_t F^t\right\| \ge \sqrt{\frac{8\log(n_1+n_2)\nu}{m_0}}\right) \le 1/(n_1+n_2).$$

A hidden condition when the above inequality holds is that $\nu$ dominates the denominator of the exponential term. The remaining work is to have sufficiently large $m_0$ to guarantee that $\nu$ dominates the denominator of the exponential, which follows

$$\nu \ge 2/3L\sqrt{\frac{8\log(n_1+n_2)\nu}{m_0}}.$$

The above inequality immediately implies that

$$m_0 \ge \frac{32}{45}n_{\min}\log(d(n_1+n_2)n_1 n_2)\log(n_1+n_2)\frac{\max_t w_t^2\left((d-t)\frac{\mu_0^2 r}{n_1}\sigma_2^2 + \sigma_1^2\right)}{\sum_{t=1}^d w_t^2\left((d-t)\sigma_2^2 + \sigma_1^2\right)}.$$

Note that $n_1 + n_2 > n_i, i = 1, 2$, and $n_1 + n_2 > d$, then the above sample complexity can be simplified as

$$m_0 \ge \frac{128}{45}n_{\min}\log^2(n_1+n_2)\frac{\max_t w_t^2\left((d-t)\frac{\mu_0^2 r}{n_1}\sigma_2^2 + \sigma_1^2\right)}{\sum_{t=1}^d w_t^2\left((d-t)\sigma_2^2 + \sigma_1^2\right)}. \tag{35}$$

The remaining work is to bound $\left\| \sum_{t=1}^{d} w_t F^t \right\|$. First we note that each entry of $F^t$ is Gaussian and the variance is not greater than $\sigma_1^2 + (d-t)\sigma_2^2$. Then, according to results on bounds for the spectral norm of i.i.d. Gaussian ensemble, we have

$$\mathbb{P}\left( \left\| \sum_{t=1}^{d} w_t F^t \right\| \leq 2\sqrt{\sum_{t=1}^{d} w_t^2 \left(\sigma_1^2 + (d-t)\sigma_2^2\right)} \sqrt{n_{\max}} \right) \geq 1 - C_1 \exp(-c_2 n_{\max}), \qquad (36)$$

where $C_1, c_2$ are absolute positive constants. Note that $C_1 \exp(-c_2 n_{\max}) \ll d n_{\max} \exp(-n_{\min})$.

Now we are ready to bound $\|J\|_2^2$. With probability at least $1 - \frac{3}{n_1 + n_2} - 5 d n_{\max} \exp(-n_{\min})$ we have

$$\|J\|_2^2 \leq p^2 \left( \left\| \sum_{t=1}^{d} w_t F^t \right\| + \sqrt{\frac{8\log(n_1 + n_2)\nu}{m_0}} \right)^2$$

$$\leq 320 p^2 \max\{n_1 n_2 \log(n_1 + n_2)/m_0, 1\} n_{\max} \sum_{t=1}^{d} w_t^2 ((d-t)\sigma_2^2 + \sigma_1^2) \qquad (37)$$

$$= 320 p^2 \sum_{t=1}^{d} w_t^2 ((d-t)w_2^2 + \sigma_1^2) n_1 n_2 \log(n_1 + n_2) n_{\max}/m_0$$

$$= \frac{320 m_0 \log(n_1 + n_2) \sum_{t=1}^{d} w_t^2 ((d-t)\sigma_2^2 + \sigma_1^2)}{n_{\min}}.$$

The first equality uses the fact that $m_0 < n_1 n_2 \log(n_1 + n_2)$.

Combining (26),(27) and (37) yields the second part of inequality (11) in Theorem 3.8.

$\square$

# F Proof of Lemma E.1

*Proof.* The proof is almost the same as the proof of Lemma 12 in [15] with some minor modifications.

Set $\mathcal{F} = \frac{44 r n_1 n_2}{m_0} \left(\mathbb{E}(\|\Sigma_R\|)\right)^2$. We will show that the probability of the following bad event is small:

$$\mathcal{B} = \left\{ \exists X \in \mathcal{E}(r) \text{ such that } \left| \sum_{t=1}^{d} w_t \left\| \mathcal{A}^t (X) \right\|_2^2 - p \|X\|_F^2 \right| > \frac{p}{2} \|X\|_F^2 + \mathcal{F} \right\}.$$

Note that $\mathcal{B}$ contains the complement of the event in Lemma E.1.

We use a peeling argument to bound the probability of $\mathcal{B}$. Let $\nu = \sqrt{\frac{2048 \sum_{t=1}^{d} w_t^2 \log(n_1 + n_2)}{\log(6/5)m_0}}$ and $\alpha = 6/5$. For $l \in \mathcal{N}$ let

$$S_l = \left\{ X \in \mathcal{E}(r) : \nu\alpha^{l-1} \leq \frac{1}{n_1 n_2} \|X\|_F^2 \leq \nu\alpha^l \right\}.$$

Then if event $\mathcal{B}$ holds for some $X \in \mathcal{E}(r)$, it must be that $X$ belongs to some $S_l$ and

$$\left| \sum_{t=1}^{d} w_t \left\| \mathcal{A}^t (X) \right\|_2^2 - p \|X\|_F^2 \right| > \frac{p}{2} \|X\|_F^2 + \mathcal{F} > \frac{5}{12} \alpha^l \nu m_0 + \mathcal{F}. \qquad (38)$$

For $T > \nu$ consider the set

$$\mathcal{E}(r, T) = \left\{ X \in \mathcal{E}(r) : \|X\|_F^2 \leq n_1 n_2 T \right\}$$

and the event

$$\mathcal{B}_l = \left\{ \exists X \in \mathcal{E}(r, \alpha^l \nu) \text{ such that } \left| \sum_{t=1}^{d} w_t \left\| \mathcal{A}^t (X) \right\|_2^2 - p \|X\|_F^2 \right| > \frac{5}{12} \alpha^l \nu m_0 + \mathcal{F} \right\}. \qquad (39)$$

Note that $X \in S_l$ implies that $X \in \mathcal{E}(r, \alpha^l \nu)$. Then (38) implies that $\mathcal{B}_l$ holds and $\mathcal{B} \subset \cup \mathcal{B}_l$. Thus, it is sufficient to bound the probability of the simpler event $\mathcal{B}_l$ and then apply the union bound. Such a bound is given by the following lemma. Its proof is given in Appendix G. Let

$$H_T = \sup_{X \in \mathcal{E}(r,T)} \left| \sum_{t=1}^{d} w_t \left\| \mathcal{A}^t (X) \right\|_2^2 - p \left\| X \right\|_F^2 \right|.$$

**Lemma F.1.** *Suppose all $\mathcal{A}^t$'s are fixed uniform sampling ensembles. Then*

$$\mathbb{P} \left( H_T > \frac{5}{12} \alpha^l \nu m_0 + \mathcal{F} \right) \leq \exp \left( \frac{-c_5 m_0 T^2}{\sum_{t=1}^{d} w_t^2} \right),$$

*where $c_5 = 1/4096$.*

The above lemma implies that $\mathbb{P}(\mathcal{B}_l) \leq \exp(-c_5 m_0 \alpha^{2l} \nu^2)$. By a union bound, we have

$$\mathbb{P}(\mathcal{B}) \leq \sum_{l=1}^{\infty} \mathbb{P}(\mathcal{B}_l) \leq \sum_{l=1}^{\infty} \exp \left( \frac{-c_5 m_0 \alpha^{2l} \nu^2}{\sum_{t=1}^{d} w_t^2} \right) \leq \sum_{l=1}^{\infty} \exp \left( \frac{-(2c_5 m_0 \log(\alpha)\nu^2)l}{\sum_{t=1}^{d} w_t^2} \right),$$

where the last inequality uses the bound $e^x \geq x$. Then, substituting $v = \sqrt{\frac{2048 \sum_{t=1}^{d} w_t^2 \log(n_1 + n_2)}{\log(6/5)m_0}}$ into the above summation we obtain

$$\mathbb{P}(\mathcal{B}) \leq 2/(n_1 + n_2).$$

This completes the proof. $\qquad \square$

## G  Proof of Lemma F.1

*Proof.* The proof is almost the same as the proof of Lemma 14 in [15] with some minor modifications.

By Massart's concentration inequality (see, e.g., [2], Theorem 14.2), we have

$$\mathbb{P} \left( H_T \geq \mathbb{E}(H_T) + \frac{1}{9} \frac{5}{12} m_0 T \right) \leq \exp \left( \frac{-c_5 m_0 T^2}{\sum_{t=1}^{d} w_t^2} \right), \tag{40}$$

where $c_5 = 1/4096$. Next we bound the expectation $\mathbb{E}(H_T)$. Using a symmetrization argument we obtain

$$\mathbb{E}(H_T) \leq 2 \mathbb{E} \left( \sup_{X \in \mathcal{E}(r,T)} \left| \sum_{t=1}^{d} w_t \gamma_i^t \sum_{i=1}^{m_0} \left\langle A_i^t, X \right\rangle^2 \right| \right),$$

where $\gamma_i^t$ is a Rademacher variable (independent on both $i$ and $t$). The assumption $\|X\|_\infty = 1$ implies that $|\langle A_i^t, X \rangle| \leq 1$. Then the contraction inequality yields

$$\mathbb{E}(H_T) \leq 8 \mathbb{E} \left( \sup_{X \in \mathcal{E}(r,T)} \left| \sum_{t=1}^{d} w_t \gamma_i^t \sum_{i=1}^{m_0} \left\langle A_i^t, X \right\rangle \right| \right) = 8 \mathbb{E} \left( \sup_{X \in \mathcal{E}(r,T)} \left| \left\langle \Sigma_R, X \right\rangle \right| \right),$$

where $\Sigma_R = \sum_{t=1}^{d} \sum_{i=1}^{m_0} w_t \gamma_i^t A_i^t$. Since $X \in \mathcal{E}(r, T)$, we have

$$\|X\|_* \leq \sqrt{r} \|X\|_F \leq \sqrt{r n_1 n_2 T}.$$

Then by the trace duality inequality, we obtain

$$\mathbb{E}(H_T) \leq 8 \sqrt{r n_1 n_2 T} \mathbb{E} \|\Sigma_R\|_2.$$

Finally using

$$\frac{1}{9} \frac{5}{12} m_0 T + 8 \sqrt{r n_1 n_2 m_0 T} \frac{1}{\sqrt{m_0}} \mathbb{E} \|\Sigma_R\|_2 \leq \frac{1}{9} \frac{5}{12} m_0 T + \frac{8}{9} \frac{5}{12} m_0 T + \frac{44 r n_1 n_2}{m_0} \left( \mathbb{E} \|\Sigma_R\|_2 \right)^2$$

combined with (40) we complete the proof. $\qquad \square$