[Reviews · NeurIPS 2016]

Reviewer 1

Summary

This paper looks at low-rank recovery problems from a few linear measurements. This is of course a problem studied at length in the current literature. The main focus of this paper however is on understanding of the effect of time evolving low-rank matrices. The hope is that for matrices that evolve slowly over time one could get a way with less measurements. The authors show that indeed this is the case.

Qualitative Assessment

comments: - Line 34 This is of course not important but the list is not really comprehensive - Lines 100-102: In principal U^t could also change e.g. movies could be added in. Overtime certain actors/actresses may become less popular etc. But I agree that these changes are much slower so that assuming U is fixed is not a bad assumption - Line 108-109 It would be interesting if the authors studied more general dynamic models -Lines 191-196: Its a bit odd that both incoherence and infinity bound (a.k.a. spikiness) is needed as the infinity bound should be enough for getting RIP. - Lines 191-196: What happens if there is no noise. (8) seems to suggest the error does not go to zero which is a bit weird. This is of course common when using spikiness model but I expect with just incoherence and without this spikiness assumption the error can be equal to zero (in the absence of noise). This I would say is the main weakness of the paper.

Confidence in this Review

3-Expert (read the paper in detail, know the area, quite certain of my opinion)


Reviewer 2

Summary

The paper studied the matrix recovery problem with two particular instances: matrix sensing and matrix completion. More concretely, a matrix $\mathcal{X}^d$ is recovered from measurements $\mathcal{Y}^t=\mathcal{A}^t(\mathcal{X})$, $t = 1, \ldots, d$, by the following scheme \begin{equation*} \hat{\mathcal{X}^d}\in \underset{\mathcal{X}}{\operatorname{argmin}}\;\sum_{t=1}^d\omega_t\|\mathcal{A}^t(\mathcal{X})-\mathcal{Y}^t\|^2, \end{equation*} where $(\omega_t)_{1\leq t\leq d}$ are chosen weights. This paper considers the model when these measurement are dependent, i.e., \begin{equation*} \mathcal{Y}^t = \mathcal{A}^t(\mathcal{X}^t)\quad\text{with}\quad \mathcal{X}^t= f(\mathcal{X}^1, \ldots, \mathcal{X}^{t-1}) + \text{noise}, \end{equation*} and provides some theoretical error bounds under two main assumptions: low-rankness and Markov property. Two recovery error bounds are proved under the assumption that either $(\mathcal{A}^t)_{1\leq t\leq d}$ are Gaussian measurement ensembles or they are uniform sampling ensembles. Alternating Direction of Multipliers Method is then used to solve optimization problem.

Qualitative Assessment

This paper provided a novel setting for matrix recovery problem by using models (2) and (7). It is better to use model (7) instead of (2) in general and state two theorem separately for two cases $a<\infty$ and $a=\infty$. Good interpretation of bounds and the choice of weights but it would be easier to follow if the authors indicate the precise results of [5] and [15] (which theorems in these references) in comparisons.

Confidence in this Review

2-Confident (read it all; understood it all reasonably well)


Reviewer 3

Summary

The paper studies a dynamic variant of the usual matrix sensing and matrix recovery problems. There is a true matrix time series that evolves in time according to a random-walk type process. At each time point, one observes a noisy and incomplete observation of the true matrix at time t. The goal is to recover the true matrix at time t using all the observations up to time t. The authors show that their approach can be much more accurate compared to the static matrix recovery approach where one uses only the observation at time t to recover the true matrix at time t.

Qualitative Assessment

I found the paper well-written and interesting. They make a good point (that dynamic recovery can be much more accurate than static recovery methods) although their theoretical results seem like straightforward extensions of the corresponding static results. One potential downside of the proposed method is that it involves too many tuning parameters (the weights w1, ..., wd, the maximum of the true matrix a for matrix completion, the unknown rank of the true matrix r etc). The authors have not described clearly a method for choosing these in practice. They say something about cross validation in their experiments section but it will be nice to make a clearer specification in the section where they introduce LOWEMS. The fact that the practitioner needs to make so many choices might limit the potential application of their method. Theoretically, of course, the downside is that the method relies on a non-convex problem so the guarantees given are for a non-computable estimator. No guarantees are given for the computable estimator based on alternating minimization.

Confidence in this Review

2-Confident (read it all; understood it all reasonably well)


Reviewer 4

Summary

The authors analyze low rank matrix estimation in a simple dynamic setting where one of the low rank factors varies smoothly across time intervals. This a first attempt at theoretical understanding of dynamic models in high dimensional matrix estimation. The authors derive consistent error bounds for the proposed dynamic model using a weighted squared error minimizer under a non-convex rank constraint.

Qualitative Assessment

My primary concerns are listed below: A. Proof 1. a) In line 379 (supplementary material) Theorem 2.3 from 5 cannot be directly used to establish RIP for \sum_t \sqrt{wt} At(\Delta) as \sum_t wt ||At(\Delta)||_2^2 NOT = ||\sum_t \sqrt{wt} At(\Delta)||_2^2. However, as theorem 3.1 requires bound on only the former, the proof in [5] can be extended. The proof needs some work though. b) Lemma B.2 inequality is reversed. B. Theory 2. The results are derived for exact global solution to (2) which is a non-convex optimization. The paper is incomplete without the suggested future work on analyzing alternating minimization. I believe the analysis will follow through with a bit of linear algebra machinery, but the exact expression of the joint error term arising from Thm 3.4 and 3.8 and the weighted power iteration is more useful towards understanding the tradeoff of sample complexity and accuracy using the dynamic estimation in practical implementations. 3. For the proposed estimator, simple extensions to the dynamic model analyzed (in line 108) can drastically increase the number of parameters to be estimated using cross validation as explained below: a) Since the e^t are all assumed to be Gaussian with mean zero and a common standard deviation of sigma_2 (independent of t). This setting essentially boils down to the standard matrix estimation problem, where the samples have different noise levels, i.e. each y^t as samples from X^d with Gaussian noise of sigma_1+\|U\|(d-t)sigma_2. b) At a high level, the optimum w^t essentially would weight the fit to the samples based on the noise level at y^t.The algorithm for computing optimum w^t requires estimating a \kappa parameter (which depends on sigma_1,sigma_2) using cross validation. c) Even for a simple extension of having different sigma_t for e^t, the optimum estimation of w^t would require essentially estimating all the different noise variances using cross validation which can be computationally demanding. d) As I understand, in a naive implementation the optimum w^t require knowledge of noise level of y^t (which is restrictive). A different estimator to avoid this dependence on noise variant might be more useful here. The following work might be relevant in this regard: Klopp, Olga, and Stéphane Gaiffas. "High dimensional matrix estimation with unknown variance of the noise." (2015). C. Experiments: What does it mean to get results at a particular \kappa? The following comments assume that the authors solve for w^t using the QP in eq 5 using \kappa=\kappa in plots. 4. Figure 1: a) I am not able to judge the significance of the differences from baseline in both experiments. Since the authors have already run the experiments 5 times, I suggest they add error bars to the Figure 1. b) Optimum \kappa for Fig 1a does not match the theoretical values for matrix sensing even for data simulated from modeling assumptions. c) It is very difficult to understand this figure and the corresponding text. The key point I think is that the optimum \kappa for min recovery error increases as sigma 1 increases. This can be highlighted by using labels to indicate the optimum \kappa as (LOWEMS), \kappa\sim1e-2 as baseline 2 and \kappa > 10e2 as baseline 1. 5. Figure 3a: Since both gamma and kappa are chosen from cross validation. The result does not necessarily indicate that the performance improvement is from choosing kappa optimally. I suggest controlling for changes in gamma in this experiment. (suggestion) 6. Figure 2 highlights the trade-off between sample complexity and accuracy achieved by LOWEMS. However, I think a better plot to illustrate the effect would be to plot two figures (a) for each model m plot the sigma 2 vs the # of samples to for error convergence (say # samples so that the error is below 1.1*min_error(m) ), and (b) for each model m plot the sigma 2 vs the #min_error(m) MINOR: Line 24: "most popular approach in recent years has focused on the use of nuclear norm minimization as a convex surrogate": This is true for theoretical analysis. The low rank constraint and alternating minimization are more popular in practice. Line 50: use Big O notation or state that C^i are constants. Line 108: equation number Thm 3.4: State that D_1 is also a constant and \delta_1,\delta_2r are RIP constants from Def 3.3 not any positive constants [EDIT]: I have read the author response and my review remains unchanged. In summary, there are some interesting ideas and potential reference value in being the first analysis of theoretical properties of a dynamic matrix completion model. However, the model analyzed is too simplistic, and the significance of the paper is further undermined by the lack of analysis for a working algorithm (like alternating minimization). I am currently unable to judge the significance of empirical improvements without the error bars (although the authors response claims to include them).

Confidence in this Review

3-Expert (read the paper in detail, know the area, quite certain of my opinion)


Reviewer 5

Summary

This paper discusses an interesting problem of approximately recovering an underlying matrix evolving over time. The paper establishes error bound for the proposed model in both the matrix sensing and matrix completion scenarios. The analysis is complete and the experiments are sufficient.

Qualitative Assessment

The paper is well written and the result looks promising. The followings are my concerns on the paper: 1. The authors do not define D_1 in Theorem 3.4, which is critical. I guess here D_1 denotes an absolute constant. If sigma_1=sigma_2=0, then Theorem 3.4 argues that the proposed model exactly recovers the underlying matrix, if the sample complexity is O(rn) (set w_t=1/d for all t). 2. Are the error bounds in Theorem 3.4 and Theorem 3.8 tight? It would be better if a lower bound could be given. 3. I would suggest the authors state how the result of Theorem 3.1 leads to Theorem 3.4 and Theorem 3.8 explicitly.

Confidence in this Review

2-Confident (read it all; understood it all reasonably well)


Reviewer 6

Summary

This papers deals with the problems of matrix completions and matrix sensing when some temporal depency is taken into account. They propose a locally weighted low rank estimator and provide an upperbound for the reconstruction error.

Qualitative Assessment

General comments - The paper is clearly written and easy to follow. However, the presentation is slightly misleading. In introduction and section 2, the authors start presenting the problem as a general dynamical problem but then end up focusing on the random walk dynamic which is very specific. - In the supplementary, line 476 why can non-commutative Bernstein inequality be applied? The matrices appearing in the sum are not necessarily independent. Indeed the random variables $h^t$ for $t =1..d$ defined in (3) are not independent (if an index $(i,j)$ is measured twice at two different dates $s$ and $t$, then $X_{ij}^s$ and $X_{ij}^t$ are correlated through the random walk noise $\epsilon_t$). Hence the sum appearing line 442 is not a sum of independent matrices. If Bernstein cannot be used then Theorem 3.8 is not proved. - The dynamic proposed in the paper is not stationary which hurts the asymptotic consistency of the results. Since the matrix process follows a random walk, its entries are not bounded as $t$ tends to infinity. Hence in Theorem 3.8 $a$ tends to infinity and so does the upper bound. - The choice of truncating Netflix dataset is very questionable, indeed a 0.74 visible entry ratio is very high (less than 0.001 in the original dataset). It would be interesting to know the results without the truncation.

Confidence in this Review

2-Confident (read it all; understood it all reasonably well)